# Addressing Delusions in Women and Men with Delusional Disorder: Key Points for Clinical Management

**DOI:** 10.3390/ijerph17124583

**Published:** 2020-06-25

**Authors:** Alexandre González-Rodríguez, Mary V. Seeman

**Affiliations:** 1Department of Mental Health, Parc Taulí University Hospital, Autonomous University of Barcelona (UAB), I3PT. Sabadell, 08280 Barcelona, Spain; agonzalezro@tauli.cat; 2Department of Psychiatry, University of Toronto, Toronto, ON M5T 1R8, Canada

**Keywords:** delusions, gender, clinical management, delusional disorder, psychosis

## Abstract

Delusional disorders (DD) are difficult conditions for health professionals to treat successfully. They are also difficult for family members to bear. The aim of this narrative review is to select from the clinical literature the psychosocial interventions that appear to work best for these conditions and to see whether similar strategies can be modeled or taught to family members so that tensions at home are reduced. Because the content of men’s and women’s delusions sometimes differ, it has been suggested that optimal interventions for the two sexes may also differ. This review explores three areas: (a) specific treatments for men and women; (b) recommended psychological approaches by health professionals, especially in early encounters with patients with DD; and (c) recommended psychoeducation for families. Findings are that there is no evidence for differentiated psychosocial treatment for men and women with delusional disorder. What is recommended in the literature is to empathically elicit the details of the content of delusions, to address the accompanying emotions rather than the logic of the presented argument, to teach self-soothing techniques, and to monitor behavior with respect to its safety. These recommendations have only been validated in individual patients and families. More rigorous clinical trials need to be conducted.

## 1. Introduction

The *Diagnostic and Statistical Manual of Mental Disorders, Fifth Edition* (DSM-5) defines delusional disorder (DD) as the presence of one or more delusions, lasting for at least one month or longer, in the absence of affective symptoms, prominent hallucinations or other symptoms of schizophrenia [1]. The prevalence of DD is estimated to be approximately 0.2% [1]. Subtypes of DD have been categorized according to the content of the primary delusion: persecutory, jealous, erotomanic, somatic, grandiose, mixed and unspecified [1,2]. Gender differences in DD have thus far been poorly studied, in marked contrast to illnesses such as schizophrenia, where symptoms in young adult women emerge later than they do in men, with a second peak of incidence at the end of the reproductive years. In schizophrenia, it has also been shown that young women respond more completely to antipsychotic medication than do men, but that this wanes after menopause. DD, in which social and personal functioning is superior to that in schizophrenia, starts later in life, which may attenuate gender difference [3]. 

For many decades, DD has been considered a difficult condition to treat, in part because both male and female patients with this diagnosis adhere poorly to prescribed medication regimens [4]. It makes sense to think that more effective clinical approaches to patients and families would enhance patient recovery even though, despite notable efforts to disseminate results of DD research promptly [5], there is still little available evidence as to what the most therapeutic approaches to DD are. It is possible that the optimal management of the two sexes is not the same [6,7].

Because gender differences have been demonstrated in a variety of psychotic illnesses, some researchers have advocated sex-specific treatment for psychosis, not only with respect to pharmacologic treatment [8,9], but also with approaches to psychosocial intervention [10].

A substantial amount of research is now available on the management of DD, but some basic aspects have remained relatively neglected. For instance, the origins of delusional beliefs, whether in men or women, continue to baffle clinicians [11] so that psychological assistance cannot easily be directed at the putative source of whatever cognitive distortions exist. These same difficulties perplex the families of patients. The seemingly unprovoked emergence of implausible beliefs in their relative bewilders them, and they are at a loss as to how best to respond [12]. Psychoeducational guidelines have been developed for families of patients with bipolar disorder [13] and schizophrenia [14], but not yet for families of sufferers from DD.

### 1.1. Method

We searched the Google Scholar and PubMed databases for English and Spanish language papers that referred to gender and the management of delusions in patients with DD. Both authors scanned the abstracts of several hundred titles; most were excluded because they addressed delusions and gender in the context of other psychoses. As there were few papers that dealt specifically with gender and the management of delusions in this condition, we expanded our search to include therapeutic approaches to delusions in general, based both on research studies and on reports and reviews of clinical experience. Where appropriate, we included results from our own clinical experience. In the end, we elected to focus on 44 articles that were most relevant to our aims. 

### 1.2. Aims

The aim of this narrative, non-systematic but critical review is to address three questions: (a) Should treatment recommendations for DD differ depending on the patient’s gender?; (b) regardless of gender, how should psychiatrists and psychologists address delusions, especially during the formative phases of treatment?; and (c) regardless of gender, how should family members respond to manifestations of delusional beliefs?

## 2. Differential Treatment of Men and Women with DD 

The vast majority of the literature indicates that DD is more prevalent in women than in men [6]. Moreover, when compared to men, women show a delayed onset of symptoms and a poorer response to available treatment. DD is mainly a disease of older age and, as women age, they require increasingly higher doses of antipsychotic drugs to keep symptoms at bay [7]. With respect to seeking assistance, women are quicker to realize that they need help, not only for schizophrenia but also for DD [8].

Male/female differences with respect to any psychiatric illness, however, depend heavily on the traditions of the local community and the values of the surrounding culture. In a 1999 study from Taiwan (86 outpatients with DD), participants were divided into DD subtypes, the most common in this study being the persecutory type. In the total group, there were more men than women. The mean age was 42.4 + 15.1 years, with women being older than men. Almost half of the group had prominent depressive symptoms but, contrary to expectations, the likelihood of depression did not differ between men and women [15].

Examining the results of studies from around the world, González-Rodríguez et al. [16] reported no significant differences between men and women in the content of their delusions. Exceptions, of course, exist. Delusions of pregnancy [17] and delusional denial of pregnancy [18] are women’s delusions, although, because DD is a condition of relatively older age, reproductive themes are less prominent here than in other psychoses. Koro, a delusion that one’s genitals are shrinking seen mostly in Asian countries [19], is, by contrast, the prototype of a specifically male delusion.

Delusions can be constructed together with others, or co-formed; this results in a condition referred to as folie à deux [20,21]. There is a lead person (the inducer) and a follower (the inductee) and the two are usually in a close relationship. The more dominant partner is the inducer, sometimes man or a woman. 

In the context of one form of DD, delusional infestation, substance abuse has been found to be significantly more prevalent in men than in women [22]. The extra male prevalence of comorbid abuse of alcohol and drugs probably holds true across other delusional states as well.

### 2.1. Pharmacotherapeutic Differences

As to treatment differences, men and women respond differently to antipsychotic medications both in terms of efficacy and in their susceptibility to adverse effects [23], but the evidence mostly comes from schizophrenia studies. In schizophrenia, women are prescribed clozapine or long-acting antipsychotics less frequently than men [24], but these drugs are not often used for DD. Again, in the context of schizophrenia, women are more likely than men to be prescribed an antidepressant, mood stabilizer, anxiolytic and sedative along with their antipsychotic [24]. Because of estrogen effects on the enzymes that metabolize olanzapine and clozapine, premenopausal women need lower doses than men at least for these two drugs [23]. This may not apply to women with DD, who are usually close to or beyond the menopause before symptoms set in [25]. In addition to age, drug kinetics are affected by many other variables, namely illness severity, comorbidities, smoking status and diet. It is also important to remember that antipsychotics are lipophilic drugs. They accumulate in fat stores and women’s bodies are composed of more adipose tissue than men’s bodies. For this reason, when antipsychotic drugs are discontinued, men are quicker to relapse. For the same reason, women who precipitously lose weight are flooded with a stored drug and suddenly suffer new adverse effects [23].

Even when weight is kept steady, some side effects of antipsychotics, such as metabolic disturbances, are more pronounced in women than they are in men, partly because the average woman takes a greater variety of drugs than the average man and is, therefore, more exposed to potential drug interactions [26]. Immunological differences between men and women also play a role. Almost all drug allergies are two to three times more prevalent among women than men [27]. 

The adverse effects of drugs are proportionally more important in DD than in other psychotic conditions because of the relatively older average age of DD patients, so that antipsychotic doses need to be kept low.

### 2.2. Psychosocial Differences

There are also psychosocial differences between men and women that influence the success of treatment. In general, women are socioeconomically disadvantaged relative to men—they may live in poorer and less hygienic surroundings, have transportation difficulties coming to psychiatric appointments, or are unable to afford healthy diets [23]. They are more likely to suffer from comorbidities and more likely to be caring for others [23], whether children or elderly relatives. If they are mothers, their delusions may affect, even endanger, their children. Women live longer than men and it is not yet fully known what old age and bereavements do to the severity of delusions. What, for instance happens to somatic delusions when severe somatic illness sets in with age? What happens to delusional jealousy of one’s spouse when the spouse dies? These are as yet unanswered questions.

There are many theoretical reasons why treatment of DD could necessitate different psychological approaches to men and women. Differences in male/female psychotherapeutic strategies have been advocated in the past [28], with an emphasis on avoiding the temptation to stereotype. As an example, it is unjust and unwise to assume that aggression is a male trait and that women with psychosis are less prone to violence than are men [29].

While sexually dimorphic non-pharmacologic guidelines for the treatment of men and women with DD may emerge in time, for the time being, the findings and recommendations cited in the rest of this paper are reported as applying equally to both sexes.

## 3. Therapeutic Conversations with Patients Suffering from Delusions 

Zangrilli and co-workers [30] qualitatively analyzed audio-recorded meetings between five psychiatrists and 14 of their patients in an acute care inpatient setting. Using content analysis, they identified six subthemes in the conversations, with three approaches achieving prominence. The most commonly occurring approach by psychiatrists was to elicit the content of the delusion and the evidence for it (subtheme 1—e.g., “What makes you believe this is so?”). The second most common approach (subthemes 2–4) was an attempt to go deeper by identifying associated emotions, exploring the connections between thought and behavior, and discussing reasons for hospital admission—e.g., “How does this makes you feel, what have you done about it, what led to your hospitalization?”. The third approach was challenging patients’ logic and suggesting alternatives to their explanations of events (subthemes 5 and 6—e.g., “Could it have been a coincidence? Could it have meant something else?”). In the study, this latter strategy often elicited defensiveness on the part of the patient. Potential alternative approaches, such as explaining physical symptoms as sequelae of a psychological disturbance—e.g., “The stress that you describe as a result of your divorce can sometimes cause breathing difficulties”—or linking symptoms to a previous history of adversity—e.g., “Your father’s betrayal must make it difficult to imagine that men can ever be trusted”—did not emerge as themes in the Zangrilli et al. study [30] (Table 1).

### 3.1. Initial Introductions

How psychiatrists initially engage with patients is generally acknowledged as important to the quality of the therapeutic relationship that subsequently develops. Priebe and collaborators [31] carried out a study on patients’ preferences for how psychiatrists introduce themselves during the first psychiatric consultation. They found that most patients preferred a lead-in that summarized what the interview would consist of, what would be asked, and what was expected of the patient. They preferred this over the psychiatrist talking about his/her own background, education, beliefs, or philosophy of treatment. This is helpful to know.

### 3.2. Responding to Delusions

How one addresses the issue of delusions in an initial consultation not only helps to establish an alliance between patient and doctor but also helps with diagnosis and may determine whether or not patients will adhere to recommended treatment [31,32]. It has been said that initial interviews, when properly conducted, can be therapeutic in themselves and can have long range effects on ultimate outcomes [32]. The nature and severity of symptoms do, of course, play a role in how an initial interview is best shaped. Federico et al. [33] conducted a study of 27 psychiatrists and 100 patients with schizophrenia investigating whether symptom levels influenced what was said during consultations. In the 27% of cases where patients brought up their delusions, 18% of psychiatrists deliberately avoided probing the issue, while 15% engaged with it. Psychiatrists were most likely to address negative and general symptoms of psychosis rather than the delusions themselves, and more likely to do so when the symptom level was relatively low. In other words, acute psychotic symptoms were not initially addressed, presumably to avoid confrontation. 

### 3.3. Establishing Trust 

Establishing trust before addressing delusions is important in order to set the stage for a firmer therapeutic alliance. People with DD tend to be suspicious, particularly if they have had prior negative experiences with physicians. Before unburdening themselves, they need to feel they will not be hurt or harmed. Active listening on the part of the psychiatrist is a good initial strategy. Patients obsessed with a delusion will talk about it once they feel safe. The first task, therefore, is to create a safe place where the patient can express fears and concerns without their conclusions being disparaged. An unhurried attitude and genuine interest in the patient’s story convey a willingness to listen and to be of help [34].

### 3.4. Empathizing with Feelings

Frequently, what a DD patient wants most from a psychiatrist is to be cleared of the insanity label intimated by family or friends. The ability to frame the delusional belief in a normative way, as an idea not necessarily out of the ordinary [35,36] is conveyed via facial expression and gesture; this goes a long way toward consolidating a therapeutic alliance. Neutrality as to the accuracy of the belief needs to be maintained, but respect for patients and empathy toward their distress can be openly expressed. When, for example, a patient suffering from a delusion of theft accuses nursing staff of stealing, one can empathize with the patient about how terrible it is to not find something one wants without agreeing that the nurse is a thief [37]. Arguing about the truth of delusional experiences is always futile. That is not to say that psychiatrists should pretend to agree with the patient; the important thing is to always show respect towards the patient’s perspective.

### 3.5. Working Together

Whatever the interviewer thinks of the logic behind the belief, it remains real to the patient. Its personal meaning is important to explore, as well outlined in a paper dealing with delusional infestation [38]. The authors recommend a shared understanding of the life factors that have contributed to the patient’s intense concern about her skin. They recommend starting with a focus on the emotional and physiological sequelae of the belief, rather than on its content. The next step they suggest is teaching distress tolerance and coping skills, and self-soothing techniques as a way to decrease panic. A further recommendation is assisting the patient in coming out of self-imposed isolation, participating in enjoyable activities, and finding purpose and meaning in life despite ongoing skin symptoms. Such steps may not dissolve a delusion, but they will strengthen the therapeutic alliance and improve the patient’s quality of life. 

### 3.6. Importance of Form Over Content

Patients will ask awkward questions. “Do you believe me?” “Can you see what I see, the crawling creatures under my skin?” Whatever one thinks of the circuitous route by which patients reach their delusional explanation of events, one can always adamantly answer, “I can certainly see your distress.” As in all psychotherapeutic exchange, it is important to identify the patient’s feelings. Simultaneously, the interviewer gauges the intensity of the conviction, the frequency of events that trigger the delusion, the duration (when the delusion first started) and how it started. Periodicity, chronicity, and the association of delusional beliefs with specific people or circumstances all contribute to the psychological formulation around which a therapeutic plan can be organized. For instance, one of our patients, a college professor, whenever she saw the color red [39] was tormented by the idea that strangers were trying to lure her into sex. Kurt Schneider (of Schneiderian first-rank symptom fame) believed that it was the form, not the content, of a delusion that permits accurate diagnosis. He observed that psychotic delusions always tended towards self-reference and were experienced by the patient as “momentous, urgent, somehow filled with personal significance as if they were signs or messages from another world” [40] (p. 33). This was true for the patient just described; red had momentous meaning for her, even though what that meaning was remained unclear.

The shared formulation of the critical life story precedents that patient and physician work towards in DD need not be historically accurate [41]. It is not the accuracy that matters but the fact of, together, being able to develop a trusting partnership. On that foundation, a successful treatment plan can gradually be co-constructed.

### 3.7. Delusions Serve Psychological Purposes

Treatment efficacy is assisted by the psychiatrist’s appreciation of the purpose that a delusion serves in an individual’s life. How does a particular delusion either interfere with or facilitate everyday function? Which relationships are affected? Does the patient act on the delusions and, if so, in what way? For instance, patients suffering from delusions of jealousy tend to minutely observe their partner and actively search for evidence of infidelity. It is intensely aggravating to the partner, but the patient’s life is thereby infused with purpose [42]. Not only that but, according to reports of women wrongfully convinced of their husbands’ infidelities, jealousy inflames sexual desire and the sexual aspect of the marital relationship improves [43]. 

Another example is delusional pregnancy. This false conviction can easily be understood as a wish fulfillment for infertile or postmenopausal women. In addition, physical abuse can often be prevented if the abuser thinks his wife is pregnant [17]. 

Yet another example is the secondary gain inherent in erotomania, the conviction by a lonely isolated person of being loved by someone of importance [44]. A therapist not recognizing the role the delusion serves can lead to tragic consequences. One of our patients, for instance, after treatment with antipsychotic medication, realized that her conviction of being loved was unreal and, as a consequence, attempted suicide [44].

The secondary gain of any particular delusion is usually culturally determined and serves very different purposes in different individuals.

### 3.8. How Fixed is the Delusion?

Important information that can be gleaned from an interview is the fixity of a delusion, whether patients are able to distract themselves from it for a period of time and what it is that distracts them. It is important therapeutically to know what intervention can act as a distraction, even if only temporarily. Shared humor often serves as a good distraction. As mentioned earlier, elderly people who are losing their memory not uncommonly develop the delusion that their caregivers are stealing from them [37]. Experienced nursing home staff are usually very adept at saying something gentle or humorous that distracts and dispels the tension that could otherwise develop. 

### 3.9. Will a Delusion Translate into Action?

It is clinically important to elicit the ways patients cope with their delusions [45]. Some people cope by avoidance (in dysmorphophobia, for instance, avoiding situations where others might mock them for their supposed deformity). Some patients may cope by confrontation. A delusionally jealous woman, for instance, may deliberately harass the woman she imagines to be her husband’s lover. Patients with somatic delusions go from doctor to doctor hoping for a cure; patients with erotomania may stalk the person they think is secretly in love with them. Patients with delusional skin infestations may scratch or cut their skin in an attempt to eliminate the supposed invaders [38,46]. Angry patients can be potentially violent to themselves or others [47]. Even grandiose delusions can result in violence, usually against the self. Imagining themselves occupying exalted positions and feeling unworthy of the honor has been reported to lead sometimes to suicide [48]. Clinicians must always carefully evaluate the potential for violent behavior in DD and, if need be, hospitalize the patient involuntarily.

## 4. Helping Family Members Address Delusional Beliefs

Onwumere and colleagues [49] explored the reported experiences of family members via a phenomenological analysis of five semi-structured nursing interviews. Six themes were identified: a lack of understanding about the source and reason behind the patient’s delusional beliefs; concerns about being harmed by the irrational behavior of their relative; fears of social consequences should the relative’s delusional content become known outside the family circle; disruption of the family member’s own social relationships; and the difficulty of trying to accumulate psychological skills to better deal with the situation. 

Family reactions are important to a patient’s recovery from psychotic illness. A recent qualitative study from the UK investigated family reactions in 14 family dyads where one family member was experiencing a psychosis for the first time [50]. Some family responses, such as withdrawal, guilt, fear, and a stigmatization of mental illness, led to the avoidance of engaging mental health services in the treatment of their ill family member. Although patients with DD are older than those with first episode psychosis, the influence of family response on help-seeking is probably similar in the two conditions. Wainwright et al. [51] conducted a focus group study (four separate groups) on the experience of 23 family members who were attempting to understand and support relatives suffering from early psychosis. The common experiences of relatives are described as despair, fear of the unknown, guilt and shame, anxiety and depression, the perception of social stigma, plus feelings of loss and economic strain. The focus groups revealed four key themes. The first was “psychosis from the relative’s perspective” (bewilderment as to what was going on). The second theme was “the relative’s relationship with the mental health system”, which was often adversarial. Families felt excluded from decision making, as if they were supernumerary. The third theme, labeled “understanding,” was a continuation of the second—families felt misunderstood, unsupported and sometimes blamed for the patient’s illness. The fourth theme, “relatives coping”, was comprised of items the families had found helpful. They especially mentioned talking to others who were in the same boat and attending groups where information was shared. There was general agreement that what was needed from care providers was practical guidance, not professional “jargon”. They noted that no professional, only other group members, had ever instructed them on helpful responses they could use when their relative expressed a delusion. This is clearly a gap that mental health services need to address.

It was noted many years ago that family patterns play an important role in the development of individual delusions [52]. The phenomenon of shared psychotic disorders or folie à deux attests to the fact that the mutual elaboration of a delusion can occur among members of a family [53]. Kaffman [52] reported on a 25 patient case series in which a familial paranoid style exerted a pathogenic influence on the designated patient. Whenever this is seen clinically, whole family therapeutic approaches are indicated. Open Dialogue for Psychosis is an intervention first described in 1995 as a psychosocial approach that extends treatment to families and other members of the patient’s social network [54]. This is a collaborative approach [55] whose most significant principles, besides welcoming the participation of patients, family members, friends, and neighbors to therapeutic sessions, are flexibility and transparency. Observed interactions among family members are characterized during therapeutic sessions, but they are not labeled as pathological and no direct attempts are made to change them [54]. The goal of therapy is to stimulate dialogue among all participants so that they are able to achieve a common understanding of the situation at hand and eventually become their own agents of change. Usually used for first episode schizophrenia, Open Dialogue shows evidence of good outcome of psychosis [56].

Whether or not family members are included in therapy sessions, they will still want to know how to respond to manifestations of delusions within the family. They need training in overcoming their own burdens and discontents and learning to listen rather than argue with their ill relative. They need to learn not to do anything behind the patient’s back, but to plan all treatment cooperatively and transparently. When confronted with a delusion, they need to be trained to not pay attention to the content but to empathize with the feeling behind it. Like health care personnel, they need to be taught to maintain a neutral stance vis à vis delusions and to try to fathom the purpose the delusion may be serving in the patient’s life. Often enough it seems to serve as a protective shield against the intrusion and demands of family life. Understanding this will suggest optimal ways of responding. Families should also be taught distraction techniques. Kindness and the application of ego-boosting techniques (asking the patient for assistance in tasks, valuing their opinion, showing respect, showing appreciation publicly) can make family life tolerable and enjoyable. At the same time, family members need to recognize the signs of impending outbursts of anger or aggression in their delusional relative. They need to be instructed in exactly the steps to take should this occur, whom to call, and how to keep themselves and the patient safe from danger.

Including families in the care of the patient is an important part of the psychiatrist’s job.

## 5. Conclusions

How are we to gauge the success of the approaches discussed in this paper? The disappearance of a delusion is perhaps too ambitious a measure against which to assess the effectiveness of interventions. There are proxy measures that can be used, such as adherence to a prescribed regimen of treatment, attendance at appointments, or the speed of discharge from hospital. Neurocognitive performance, the evaluation of everyday function, and the intensity or frequency counts of delusional episodes can also be used [57]. It would be useful to evaluate treatment effectiveness in men and women separately since they may, in the future, turn out to respond best to different interventions. We need to develop operational definitions for therapeutic response in patients with DD. A recent review by González-Rodríguez and co-workers [58] examined and analyzed definitions used for antipsychotic response in DD and assessed the methodology used in studies to date. They found a general lack of consensus and a high degree of heterogeneity of the reported methods. 

At this point, it is not clear whether responses to current treatment significantly differ between men and women, but it is always wise to keep the door open to such a possibility. Optimal interviewing techniques and psychological interventions with patient and family need to be better specified. Empathically eliciting the details of the content of delusions, addressing the accompanying emotions, and monitoring behaviors seem, at present, to hold clinical validity. Instructing family members on similar strategies plus distraction and ego-boosting techniques holds promise for alleviating family distress. Investigating the influence of these approaches on outcome measures of delusional disorders suggests broad, unexplored avenues of research.

In summary, this review has focused on three main areas: (a) the possibility of differentiated treatment for men and women with delusional disorder, (b) recommendations on psychological approaches to delusional disorders, and (c) psychoeducational recommendations for families. There is no evidence that either of the latter two need to differentiate between genders. The main clinical recommendations for both professionals and family members are to empathically elicit the details of the content of delusions, to address the accompanying emotions rather than the logic/illogic of the argument, to monitor safety, and to teach the patient self-soothing techniques. While these recommendations appear valid for individual patients and individual families, they need to be tested more rigorously and on a wider scale. 

## Figures and Tables

**Table 1 ijerph-17-04583-t001:** Therapeutic approach to patients suffering from delusions.

Issues	Purpose	Recommendation
(1) Introductory remarks	To engage with patients	Summarize the steps the interview will take
(2) How to respond to a patient’s delusions	To establish alliance and help with diagnosis and treatment	When possible, avoid commenting on the factual basis of a delusion
(3) Establishing trust	To build a foundation for working together	Show genuine interest in the patient’s story
(4) Empathizing with feelings	To consolidate the therapeutic alliance	Show appreciation of the patient’s distress
(5) Working together	To create a shared goal for therapy	Focus on distress tolerance and coping skills, not on the delusion itself
(6) Importance of form over content	To identify cognitive biases	Help the patient recognize and alter modes of thinking
(7) Psychological purposes served by delusion	To understand what is gained by a delusion	Help patient recognize the role played by the delusion
(8) Fixity of delusion	To establish possibilities for distraction	Use distraction techniques when needed and teach them to family members
	To monitor safety	Carefully assess potential for self-harm and aggression

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
