# Peer review of "Addressing Delusions in Women and Men with Delusional Disorder: Key Points for Clinical Management"

_ijerph, 2020, doi:10.3390/ijerph17124583_

Round 1
Reviewer 1 Report
The paper Addressing delusions in women and men with delusional disorder: key points for clinical management by Alexandre González-Rodríguez, Mary V. Seeman, addresses a narrative analysis of delusional disorder, marked by three blocks: whether there is a differentiating treatment between men and women, the approach taken in consultations, from a psychological point of view, and psychoeducational strategies with families In the reviewed literature, it is clear that based on these three blocks, specific strategies are embroidered that can help patient recovery . Addressing feelings, establishing trust, and working together are necessary conditions for optimal patient outcomes. It seems to me a very interesting topic, which still needs to be investigated due to the variability of the responses and the actions to be taken that may favor the patient.
Two aspects to point out in the text structure of the article, two aspects to consider arise:
- With respect to the reference in Table 1, (on line 135, page 3), from the moment it is mentioned in the text until it appears (page 6), it is very separate, if it is intended to refer to what indicated.
- In line 204-207, page 5, it is not specified who makes the statement developed in the paragraph "The shared formulation of the critical precedents of the life story that the patient and the doctor work in DD does not need to be historically accurate. It is not precision that matters, but the fact that together we develop a trusted partnership. On that basis, a successful treatment plan can be gradually built.
Author Response
We are very grateful for allowing us the opportunity to revise our review entitled “Addressing delusions in women and men with delusional disorder: key points for clinical management”. We are grateful for your comments; they have helped us to improve our manuscript. All the suggestions have been addressed and we have made changes in the manuscript. Relevant changes are highlighted in yellow in the revised manuscript.
Thank you for your positive comments.
A) We have removed the first mention of Table 1 as it is, as you point out, confusing to the reader.
B) We have added reference #41 for the paragraph where it was missing.
Reviewer 2 Report
The aim of this narrative review is to select from the clinical literature the psychosocial interventions that appear to work best and to see whether similar strategies can be taught to family members so that tensions at home are reduced. This article is timely an generally well-writen paper.There are a number of questions and suggestions that arise in the reading of this paper that are reported below.
1- Although this is a narrative review from a methodological point of view, the authors should come close to the PRISMA guidelines.
2- A chapter entitled "method" should be introduced to describe how the articles were selected (search strategy, study detection, Inclusion and exclusion criteria, Data collection…)
3- Some references are very old. Are the timely?
- Hsiao, M.C.; Liu, C.Y.; Yang, Y.Y.; Yeh, E.K. Delusional disorder: Retrospective analysis of 86 Chinese outpatients. Psychiatry Clin. Neurosci. 1999, 53, 673-676.
- Silveira, J;, Seeman, M.V. Shared psychotic disorder: A critical review of the literature. Can. J. Psychiatry. 1995, 40, 389-395.
- Maher, B.A. Anomalous experience in everyday life: Its significance for psychopathology. The Monist, 1999, 82, 547-70.
- Seeman, M.V. Pathological jealousy. Psychiatry. 1979, 42, 351-361.
4- You cited a lot of review. It's usually best to cite original article with high standard of proof.
5-Chapter 2.2. for the 1st chapter no reference is cited?
6- There is a missing a "discussion" section in the structuring of the article. In this case, the "conclusion" could answer the questions formulated in the introduction.
Author Response
We are very grateful for allowing us the opportunity to revise our review entitled “Addressing delusions in women and men with delusional disorder: key points for clinical management”. We are grateful for your comments; they have helped us to improve our manuscript. All the suggestions have been addressed and we have made changes in the manuscript. Relevant changes are highlighted in yellow in the revised manuscript.
A) PRISMA guidelines were not applied as this is a narrative review, based on clinical literature, and the cumulative clinical experience of the authors, as now explained under Method.
B) Thank you for pointing out that we had omitted the Method section. We have now added a paragraph that describes the method we used to gather information.
C) It is true that several of our references are old. This is because these references are based on clinical descriptions that are often not surpassed later in time.
D) You are, of course, correct that, for reviews of research papers, it is mandatory to cite primary papers. We don’t believe that this is as compelling a need when the review covers clinical papers.
E) We have added references to Section 2.2.
F) As per your recommendation, we have now included a summary paragraph at the end of the manuscript. Thank you for this suggestion.
Reviewer 3 Report
The authors provided an interesting and useful narrative review on the clinical management of patients with delusional disorder. The contribution appears to be particularly useful for both clinicians and relatives of patients. it also offers a multifaceted rather than monolithic picture of the phenomenon, differentiating treatments and characteristics of maniefestation for male and female patients. To further improve the work I would propose the following suggestions to the authors: - line 71 et seq: the authors use the metaphor of "transmission" of delusions. However when w consider this kind of phenomena this metaphor is not the most appropriate because the psychological phenomena are always the result of circular, systemic and complex processes. I would therefore suggest to use the metaphor of co-creation or co-construction. - in the introduction a brief distinction can be made in diagnostic, prognostic, clinical, epidemiological and treatment terms between delusional disorders and the broader category of psychotic disorders. - in paragraph 3.7 and in section 4 it would be very useful to refer to a) the many systemic interpretations present in the literature of the relational and family effects that delusional disorders can have b) the interventions that systemically involve the family and the couple as the delusion was created by and is functional to the wider relational system than the individual; within these interventions I mention one that has shown particular efficacy: the open dialogue approach. - a control of the phrasing and wording in English by a native speaker would perhaps make the linguistic style more neutral and comprehensible with more immediacy.Author Response
We are very grateful to you for allowing us the opportunity to revise our review entitled “Addressing delusions in women and men with delusional disorder: key points for clinical management”. Your insightful comments have helped us to improve the manuscript. All suggestions have been addressed and we have made changes in the revised manuscript, highlighted in yellow.
A) We have changed the wording re"transmission" of a delusion.
B) We have made the introductory statements that you recommend.
C) and D) We have briefly mentioned systemic and family effects and discussed the contributions of Open Dialogue in Section 4.
E) Thank you for your comments on linguistic style. It is always difficult to know when one’s own language is biased, unclear, and when it lacks immediacy. Some changes have been made (not individually highlighted). Should it be necessary, please point out where more are needed. This is important since we pride ourselves on our language use.
Round 2
Reviewer 2 Report
I am grateful to the authors for taking my suggestions into account.